# ChiFlow: Torsional Asymmetry Flow Matching for Chirality-aware Protein Backbone Generation

## Abstract

Protein backbone generation is critical for de novo protein design, yet existing methods suffer from two key limitations: over-reliance on SE(3) modeling, which introduces unnecessary complexity for cyclic dihedral angles, and lack of explicit chirality control, leading to nonfunctional D-chiral outputs. We present **ChiFlow**, a chirality-aware backbone generator based on flow matching on **toroidal Riemannian manifolds**. ChiFlow models backbone dihedrals $\phi, \psi, \omega$ as points on $\mathbb{T}^3$, extending PPFlow to backbone variables and using periodicity to avoid boundary artifacts. Unlike the previous SE(3)-based flows such as Frameflow and Foldflow2, ChiFlow operates directly on the hypertorus, simplifying computations for angles. We also introduce a Riemannian mirroring operator and impose asymmetry on the learned vector field to enforce L-chirality. And we extended the methods in Foldingdiff by reconstructing the 3D atomic coordinates using fixed bond lengths and trigonometric calculations. To increase the diversity that was lowered by the chirality constraint, we added Stochastic Flow Matching to ChiFlow, resulting in an increase in diversity of the generated backbone. With extensive experiments on real-world protein datasets, ChiFlow approaches the leading flow models in the benchmark while maintaining absolute chirality purity. Our implement detail is at `https://anonymous.4open.science/r/anonym1`.

## 1 Introduction

Proteins play a pivotal role in sustaining life, as their structural diversity underlies a wide spectrum of biological functions. The ability to design proteins with predetermined architectures and activities represents a major advance in biotechnology, providing systematic approaches that extend beyond natural evolutionary processes. Progresses made by methods like AlphaFold3 (Abramson et al., 2024) and ESM3 (Hayes et al., 2024) carries profound significance for medicine and public health, offering the potential to generate novel therapeutic strategies and address long-standing challenges in disease prevention and treatment. At the core of protein design lies the generation of accurate backbone structures (Tang et al., 2024a), since the backbone determines the overall fold and strongly constrains the positioning of side chains. Figure 1 shows the comparasion between the whole protein and the protein backbone.

**Limitations of Existing Approaches**: Current SE(3)-based frameworks (Fuchs et al., 2020) represent protein backbones as sequences of SE(3) rigid frames (Yim et al., 2023a;b; 2024; Bose et al., 2024; Huguet et al., 2024), which are effective for capturing global structure but impose unnecessary complexity when modeling cyclic dihedral angles $\phi, \psi, \omega$—the primary determinants of backbone conformation—by embedding them into SE(3)'s rotation component, thereby creating periodicity mismatches and boundary artifacts (Zhang et al., 2024). Moreover, natural proteins are exclusively L-chiral, yet existing approaches (Yim et al., 2023a;b; 2024; Bose et al., 2024; Fu et al., 2023; Song et al., 2023) lack explicit chirality enforcement, and they mostly base on SE(3) which can not model chirality (Dumitrescu et al., 2025b; Childs et al., 2025), which risks generating non-physical right-handed backbones (Childs et al., 2025). While torus geometry has been shown to effectively capture periodicity for sidechain torsion angles (Huguet et al., 2024), it has not been extended to backbone dihedrals or coupled with chirality constraints. Therefore, there remains a critical gap for a backbone generation framework that jointly leverages torus geometry and chirality-aware modeling.

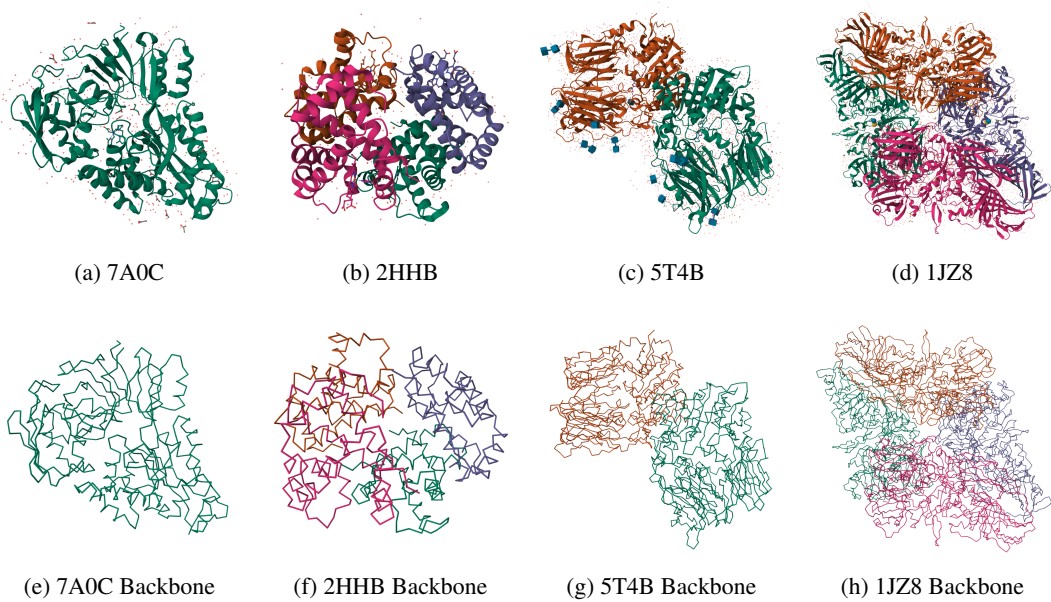

(a) 7A0C      (b) 2HHB      (c) 5T4B      (d) 1JZ8

(e) 7A0C Backbone    (f) 2HHB Backbone    (g) 5T4B Backbone    (h) 1JZ8 Backbone

Figure 1: The comparasion between the original protein and the protein backbone

**Proposed Method**: To address these limitations, we introduce **ChiFlow**, a chirality-aware flow-matching framework that operates directly on toroidal Riemannian manifolds (Chen & Lipman, 2024) rather than SE(3) flow (Bose et al., 2024). Unlike SE(3) FrameFlow (Yim et al., 2023a), ChiFlow removes the burden of embedding periodic angles into rigid-body rotations; and unlike sidechain-focused torus approache (Lin et al., 2024a), it directly targets the backbone, the core structural determinant of protein folds. Through this integration, we achieve both competitive designability and sampling efficiency, while uniquely delivering **one hundred percent chirality purity**, bridging a fundamental gap in backbone generation.

**Contributions**: ChiFlow introduces three key advances in protein backbone generation: (1)**Toroidal Riemannian Manifold** framework that models backbone dihedrals natively on $\mathbb{T}^{3N}$, eliminating periodicity artifacts inherent in SE(3)-based methods; (2) **Chirality-Aware Flow Matching** approach via a Riemannian mirroring operator and asymmetric vector field constraint, guaranteeing absolute L-chirality without sacrificing designability; and (3) **Stochastic Flow Matching** on the torus that enhances conformational diversity while preserving geometric constraints.

## 2 BACKGROUND & PRELIMINARIES

### 2.1 PROTEIN BACKBONE DIHEDRALS & CHIRALITY

A protein backbone consists of $N$ residues linked by peptide bonds (Abdin & Kim, 2023), with each residue's conformation defined by three dihedral angles : $\phi$ around the N-C$\alpha$ bond, $\psi$ around the C$\alpha$-C bond, and $\omega$ around the C-N peptide bond, which is nearly fixed at 0 in the trans state or $\pi$ in the cis state but remains cyclic. These angles determine secondary structures such as $\alpha$-helices, with $\phi \approx -57°$ and $\psi \approx -47°$, and $\beta$-sheets, with $\phi \approx -130°$ and $\psi \approx 120°$ (Ramachandran et al., 1963). And the dihedral angles will determine the chiral of the protein backbone, note that natural proteins are all inherently L-chiral (Cintas, 2002). A protein back-

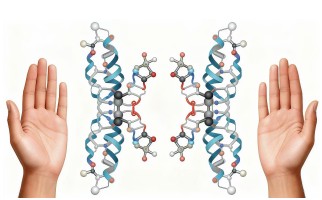

Figure 2: Chirality

bone is L-chiral if its dihedral angles, when mapped to 3D C$\alpha$ coordinates, cannot be superimposed on their mirror image, that is, the D-chiral configuration, via translations alone. To model this attribute, let $X = \{x_1, \dots, x_N\} \in \mathbb{R}^{3N}$ be C$\alpha$ coordinates derived from dihedrals $\chi \in \mathbb{T}^{3N}$, and

$M : \mathbb{R}^3 \rightarrow \mathbb{R}^3$ be a mirroring operator, for example $M(x) = (-x_1, x_2, x_3)$. $X$ is L-chiral if $\nexists b \in \mathbb{R}^3$ such that $M(X) + b = X$.

## 2.2 TOROIDAL RIEMANNIAN MANIFOLDS

The 1-dimensional torus $\mathbb{T}$ can be defined as the quotient space $\mathbb{T} = \mathbb{R}/2\pi\mathbb{Z}$ (Lipman et al., 2023), where two real numbers $a$ and $b$ are equivalent if $a - b = 2\pi k$ for some $k \in \mathbb{Z}$, making it a natural model for cyclic variables (Lin et al., 2024b) like dihedral angles. For protein backbone dihedrals, we work on the hypertorus $\mathbb{T}^3 = \mathbb{T} \times \mathbb{T} \times \mathbb{T}$, where each point $\chi = (\chi_\phi, \chi_\psi, \chi_\omega) \in \mathbb{T}^3$ represents the three dihedral angles of a residue. At a point $\chi$, the tangent space $T_\chi \mathbb{T}^3$ is endowed with a conformal product metric $g_{\mathbb{T}^3}|_\chi(u, v) = \sum_{k \in \{\phi, \psi, \omega\}} w_k(\chi_k) u_k v_k$, where the smooth, $2\pi$-periodic weights are given by $w_k(\chi_k) = \frac{1}{1 + \sin^2(\chi_k/2)}$. Extending this to $N$ residues that make up the whole protein backbone yields the product metric $g = \bigsqcup_{n=1}^{N} g_{\mathbb{T}^3}$, which induces the pointwise norm $\|u\|_g^2 = g(u, u)$ and the volume element $d\mathrm{vol}_g = \sqrt{\det g(\chi)} \, d\chi$. Under this metric, the Riemannian gradient, divergence, and Laplace–Beltrami operators are respectively $\nabla_g f = g^{-1} \cdot \partial f$, $\mathrm{div}_g X = \frac{1}{\sqrt{\det g}} \partial_i(\sqrt{\det g} \, X^i)$, and $\Delta_g f = \mathrm{div}_g(\nabla_g f)$, all computed componentwise due to the product structure. For interpolation, we follow (Zhang et al., 2024) and define the geodesic $\chi_t : [0, 1] \rightarrow \mathbb{T}^3$ between $\chi_0$ and $\chi_1$ as $\chi_t = \exp_{\chi_0}(t \log_{\chi_0}(\chi_1))$ Mathieu & Nickel (2020), which in the flat product metric reduces to shortest-path interpolation on each circle, meaning that each angular coordinate evolves linearly along the minimal arc between its start and end values, with wrapping modulo $2\pi$ to preserve periodicity.

## 2.3 FLOW MATCHING ON RIEMANNIAN MANIFOLDS

Riemannian Flow Matching (RFM) provides a simulation-free framework for learning continuous normalizing flows on general Riemannian manifolds. Given a complete, connected smooth Riemannian manifold $\mathcal{M}$ with metric $g$, the goal is to learn a time-dependent vector field $v_t(x)$ that transports an initial distribution $p_0$ to a target distribution $p_1$ by satisfying the continuity equation $\partial_t p_t + \mathrm{div}_g(p_t v_t) = 0$ with boundary conditions $p_{t=0} = p_0$ and $p_{t=1} = p_1$ Chen & Lipman (2024).

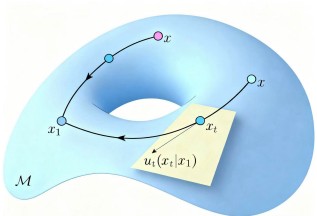

Figure 3: Torus Manifold (Chen & Lipman, 2024)

The key idea is to construct conditional probability paths $p_t(x|x_1)$ that interpolate between $p_0$ and Dirac distributions $\delta_{x_1}$ centered at data points $x_1 \sim p_1$. The marginal probability path is then defined as:

$$p_t(x) = \int_{\mathcal{M}} p_t(x|x_1) p_1(x_1) d\mathrm{vol}_{x_1}. \tag{1}$$

and the target vector field is obtained by marginalizing conditional vector fields:

$$u_t(x) = \int_{\mathcal{M}} u_t(x|x_1) \frac{p_t(x|x_1) p_1(x_1)}{p_t(x)} d\mathrm{vol}_{x_1}. \tag{2}$$

where $u_t(x|x_1)$ generates $p_t(x|x_1)$ from $p_0$.

For simple geometries where closed-form geodesics are available, the conditional flow can be defined using the geodesic distance $d_g$ as a premetric. With the linear scheduler $\kappa(t) = 1 - t$, the conditional flow is:

$$x_t = \psi_t(x_0|x_1) = \exp_{x_0}(t \log_{x_0}(x_1)). \tag{3}$$

which yields the conditional vector field:

$$u_t(x_t|x_1) = \frac{d}{dt} x_t = \frac{\log_{x_t}(x_1)}{1 - t}. \tag{4}$$

This ensures that the interpolation path follows the geodesic of the manifold, respecting its intrinsic geometry (Bose et al., 2024; Chen & Lipman, 2024).

The parametric vector field $v_\theta$ is learned by minimizing the Riemannian Conditional Flow Matching (RCFM) objective:

$$\mathcal{L}_{\text{RCFM}}(\theta) = \mathbb{E}_{t\sim\rho(t), x_1\sim p_1, x_0\sim p_0} \left[ \|v_\theta(t, x_t) - u_t(x_t|x_1)\|_g^2 \right]. \tag{5}$$

where $\rho(t)$ is a time weighting distribution, often chosen as $\rho(t) \propto (1-t)^\alpha$ for $\alpha \geq 0$, and expectations are taken over the conditional coupling induced by the geodesic interpolation (Chen & Lipman, 2024). This objective is equivalent to the marginal Riemannian Flow Matching objective up to an additive constant, ensuring that the learned vector field generates flows that satisfy the continuity equation in expectation.

RFM is simulation-free on simple geometries and does not require divergence computation during training, and provides exact target vector fields without approximation errors (Chen & Lipman, 2024). For general manifolds, spectral distances can be used as premetrics to maintain tractability, ensuring broad applicability across diverse geometries.

## 2.4 MIRRORING OPERATOR FOR CHIRALITY

To enforce strict L-chirality in generated backbones, we introduce a Riemannian mirroring operator $M : \mathbb{T}^{3N} \to \mathbb{T}^{3N}$ that acts residue-wise on the dihedral angles. For each residue, the operator is defined as:

$$M(\phi_n, \psi_n, \omega_n) = (2\pi - \phi_n \bmod 2\pi,\ 2\pi - \psi_n \bmod 2\pi,\ \omega_n). \tag{6}$$

This transformation corresponds to the molecular mirroring operation that converts an L-chiral backbone to its D-chiral enantiomer while preserving the cyclic nature of the dihedral angles. Crucially, $M$ is an isometric diffeomorphism on $\mathbb{T}^{3N}$ equipped with the product metric $g$. The pushforward $M_* : T_\chi \mathbb{T}^{3N} \to T_{M(\chi)} \mathbb{T}^{3N}$ has a constant Jacobian $J_M = \text{diag}(-1, -1, 1)$ per residue, satisfying $M^2 = I$. The conformal weights $w_k(\chi_k) = \frac{1}{1+\sin^2(\chi_k/2)}$ satisfy $w_k(\chi_k) = w_k(2\pi - \chi_k)$ for $k \in \{\phi, \psi\}$, and $w_\omega(\chi_\omega) = w_\omega(\chi_\omega)$, ensuring that $M$ preserves the metric:

$$g|_{M(\chi)}(M_* u, M_* v) = g|_\chi(u, v). \tag{7}$$

Through this, we explicitly encodes chirality awareness into the generative process, ensuring that all sampled backbones maintain correct L-chirality while preserving the natural periodicity of dihedral angles.

# 3 METHODOLOGY: CHIFLOW

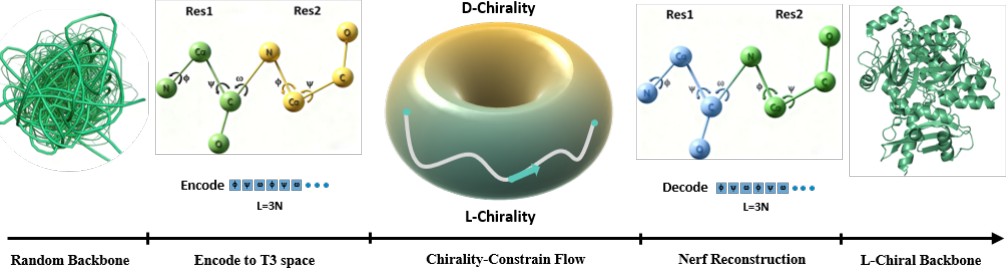

Figure 4: ChiFlow represents protein backbone dihedrals $(\phi, \psi, \omega)$ on a toroidal manifold $T^3$ to inherently preserve angular periodicity. And employs Riemannian flow matching to learn a vector field that transports noise distributions to data distributions on this manifold. To ensure strict L-chirality, a mirroring operator M is introduced, constraining the vector field to be asymmetric under M (i.e., $\hat{v}(M\chi, t) = -M_* \hat{v}(\chi, t)$). Finally, 3D atomic coordinates are reconstructed deterministically from dihedrals using fixed bond lengths and trigonometric operations via the SN-NeRF algorithm.

**ChiFlow** is a generative model based on flow-maching model for protein backbone generation, addressing critical flaws of SE(3)-based methods by grounding dihedral angle $(\phi, \psi, \omega)$ modeling in a toroidal manifold $\mathbb{T}^{3N}$ (one $\mathbb{T}^3$ per residue). Geodesic interpolation on $\mathbb{T}^{3N}$ enables continuous,

shortest-path transitions between conformations, and Riemannian operators ($\nabla_g, \text{div}_g, \Delta_g$) ensure consistent calculus on the manifold. ChiFlow first uses deterministic Riemannian flow matching to learn a vector field $\hat{v}(\chi, t)$ that transports a prior distribution $p_0$ to a native-like target $p_1$, minimizing a $L^2$ loss $\mathcal{L}_{\mathbb{T}^3}$. To enhance conformational diversity and model robustness, we extended it to Stochastic Flow Matching(SFM): a torus Brownian motion (with generator $\sigma(t)^2/2\Delta_g$, where $\sigma(t)$ is a decreasing noise schedule) defines a reference stochastic process, with noise projected onto the manifold's tangent spaces to preserve geometric validity. Framed as a Schrödinger bridge (Tang et al., 2024b), SFM learns the optimal vector field $v_\theta$ via a flow matching loss $\mathcal{L}_{\text{SFM}}$, avoiding the computational cost of simulating the entire stochastic process. Critically, to guarantee biologically essential L-chirality, a **chiral asymmetry loss** $\mathcal{L}_{\text{chiral}}$ enforces $v_\theta(M(\chi_t), t) = -M_* v_\theta(\chi_t, t)$, where $M$ is a chirality-flipping mirroring operator. To reconstruct the protein backbone using the dihedral angles generated by the flow model, we employed the SN-NeRF algorithm (Parsons et al., 2005), paired with forward kinematics using standard protein backbone bond lengths (C$\alpha$–N = 1.45 Å, C$\alpha$–C = 1.54 Å) for efficiency and stability.

### 3.1 BACKBONE MODELING AND TOROIDAL FLOW MATCHING

Proteins fold into specific three-dimensional structures to perform their biological functions (Pollock, 2007). The backbone dihedral angles—phi ($\phi$), psi ($\psi$), and omega ($\omega$)—are crucial degrees of freedom that define the protein's conformational space. Unlike linear Euclidean spaces, these angles are cyclic by nature, with values wrapping around every $2\pi$ radians (Atavin & Vilkov, 2003). This periodicity must be respected in any meaningful mathematical model of protein structure and dynamics. To address this, we turn to the concept of the Torus (Wiemeler, 2015). We represent the full protein backbone as a point on a hypertorus, $\chi = \{\chi_1, \ldots, \chi_N\} \in \mathbb{T}^{3N}$, where each residue $n$ has its own set of dihedral angles $\chi_n = (\phi_n, \psi_n, \omega_n)$. The 1-dimensional torus $\mathbb{T}$ is defined as the quotient space $\mathbb{T} = \mathbb{R}/2\pi\mathbb{Z}$, which identifies points $a$ and $b$ whenever $a - b = 2\pi k$ for any integer $k$. This identification effectively gather together the ends of the interval $[0, 2\pi)$, forming a circle and eliminating artificial boundaries at 0 and $2\pi$. The 3-dimensional product torus $\mathbb{T}^3 = \mathbb{T} \times \mathbb{T} \times \mathbb{T}$ thus naturally encodes the three dihedral angles of a single residue without introducing discontinuities. The space $\mathbb{T}^{3N}$ is not just a set of points, it is a Riemannian manifold. At a point $\chi \in \mathbb{T}^3$, the tangent space $T_\chi \mathbb{T}^3$ is a vector space containing all possible directions of change for the dihedral angles. We equip this space with a conformal product metric:

$$g_{\mathbb{T}^3}|_\chi(u, v) = \sum_{k \in \{\phi, \psi, \omega\}} w_k(\chi_k)\, u_k v_k, \quad \text{where} \quad w_k(\chi_k) = \frac{1}{1 + \sin^2(\chi_k/2)}. \tag{8}$$

This metric is smooth and $2\pi$-periodic in each coordinate, meaning it respects the circular nature of the angles. The weight function $w_k(\chi_k)$ is chosen to reflect known physical constraints; for instance, it assigns higher cost to changes in angles when they are near regions of steric clash (Ramachandran et al., 2011). Extending this metric to $N$ residues yields the product metric $g = \bigsqcup_{n=1}^N g_{\mathbb{T}^3}$ for the entire backbone. This metric induces a norm $\|u\|_g^2 = g(u, u)$ on tangent vectors and a volume element $d\text{vol}_g = \sqrt{\det g(\chi)}\, d\chi$ for integration on the manifold. On a Riemannian manifold, the familiar calculus operations from Euclidean space generalize. The Riemannian gradient, divergence, and Laplace–Beltrami (Zhou & Lähner, 2025) operators become:

$$\nabla_g f = g^{-1} \cdot \partial f, \quad \text{div}_g X = \frac{1}{\sqrt{\det g}}\, \partial_i\left(\sqrt{\det g}\, X^i\right), \quad \Delta_g f = \text{div}_g(\nabla_g f). \tag{9}$$

Due to the product structure of our torus, these operators can be computed componentwise, which significantly simplifies calculations. A key geometric concept is the *geodesic*—the shortest path between two points on the manifold. For interpolation between two backbone configurations $\chi_0$ and $\chi_1$ on $\mathbb{T}^{3N}$, we follow the shortest-path geodesic (Wu et al., 2025) on each individual angular circle:

$$\chi_t = \exp_{\chi_0}\big(t \log_{\chi_0}(\chi_1)\big). \tag{10}$$

This operation reduces to linear interpolation along the minimal arc in each angular coordinate, with values wrapped modulo $2\pi$. This provides a natural and continuous path between the desired protein backbone and the origin protein backbone. *Flow matching* is a powerful framework for learning probability paths between distributions on manifolds. On $\mathbb{T}^{3N}$, we seek a time-dependent vector field $\hat{v}(\chi, t)$ that transports an initial probability distribution $p_0$ (e.g., a simple prior distribution over

structures) to a target distribution $p_1$ (e.g., a distribution of native-like structures) by satisfying the continuity equation:

$$\partial_t p_t + \text{div}_g(p_t \, \hat{v}(\cdot, t)) = 0. \tag{11}$$

with boundary conditions $p_{t=0} = p_0$ and $p_{t=1} = p_1$. Using the geodesic interpolation $(\chi_t)_{t\in[0,1]}$ defined between samples $\chi_0 \sim p_0$ and $\chi_1 \sim p_1$, the exact conditional vector field that transports $\chi_0$ to $\chi_1$ along the geodesic is known in closed form: $u(\chi_t, t \mid \chi_1) = \frac{\log_{\chi_t}(\chi_1)}{1-t}$. We estimate the vector field $\hat{v}$ by minimizing a time-weighted Riemannian $L^2$ mismatch loss:

$$\mathcal{L}_{\mathbb{T}^3} = \mathbb{E}_{t\sim\rho} \, \mathbb{E}_{p(\chi_0), p(\chi_1)} \big[ \|u(\chi_t, t \mid \chi_1) - \hat{v}(\chi_t, t)\|_g^2 \big]. \tag{12}$$

where $\rho(t) \propto (1-t)^\alpha$ is a weighting function that often puts more emphasis on later times (e.g., $\alpha = 2$), and the inner expectation is over the conditional coupling induced by the geodesic interpolation. This geometric formulation provides a rigorous foundation for modeling protein dynamics, folding pathways (Englander & Mayne, 2014), and for generating novel structures, as it inherently avoids the coordinate singularities and artifacts that plague methods treating dihedral angles as linear Euclidean variables.

## 3.2 STOCHASTIC FLOW MATCHING AND CHIRALITY CONTROL

To address the limitations of deterministic flow and significantly improve the sample diversity and robustness of the ChiFlow model, we extend its framework by incorporating a **Stochastic Flow Matching (SFM)** (Albergo et al., 2023) formulation on the toroidal manifold $\mathbb{T}^{3N}$. This extension transforms the deterministic flow into a stochastic bridge, leveraging controlled noise and optimal transport theory.

### 3.2.1 STOCHASTIC DYNAMICS AND THE SCHRÖDINGER BRIDGE

The core of the SFM involves defining a reference stochastic process on $\mathbb{T}^{3N}$. Let $\Delta_g$ be the Laplace–Beltrami operator under the product metric. We consider a torus Brownian motion with generator $\frac{\sigma(t)^2}{2}\Delta_g$ as the reference process, where $\sigma(t)$ is a carefully chosen decreasing noise schedule. The dynamics of the system are then governed by the following Stratonovich stochastic differential equation (SDE) (Anderson, 1982):

$$d\chi_t = v_\theta(\chi_t, t) \, dt + \sigma(t) \, \Pi_{T_{\chi_t}} \circ dW_t. \tag{13}$$

Here, $\Pi_{T_{\chi_t}}$ projects the standard Wiener noise $dW_t$ onto the tangent space of the manifold at $\chi_t$, ensuring that the added noise respects the geometric constraints of $\mathbb{T}^{3N}$. The vector field $v_\theta$, parameterized by a neural network, learns to guide this stochastic process. We seek the *most efficient* stochastic path connecting the initial and target distributions relative to this reference Brownian motion. This is formulated as the Schrödinger bridge problem, an entropy-regularized optimal transport problem on the path space:

$$\min_{u_t} \, \mathbb{E} \int_0^1 \frac{\|u_t(\chi_t)\|_g^2}{2\sigma(t)^2} \, dt \quad \text{subject to} \quad \partial_t p_t = -\text{div}_g(p_t u_t) + \frac{\sigma(t)^2}{2}\Delta_g p_t. \tag{14}$$

The solution to this problem yields the control $u_t$ that minimizes the expected energy required to drive the system from $p_0$ to $p_1$ amidst the stochastic noise.

### 3.2.2 CHIRALITY PRESERVATION

Directly solving the Schrödinger bridge can be computationally challenging (Jing et al., 2025). Instead, we adopt a **flow matching** objective to efficiently learn the optimal vector field $v_\theta$ that approximates the solution. We define a conditional probability path (Mathieu & Nickel, 2020) between samples. The flow matching loss is constructed to regress the learned vector field $v_\theta$ towards a target vector field $u_\sigma$ that defines the desired flow:

$$\mathcal{L}_{\text{SFM}} = \mathbb{E}_{\chi_1 \sim p_1, \, t\sim U(0,1)} \left[ \|, u_\sigma(\chi_t, t \mid \chi_1) - v_\theta(\chi_t, t) \|_{g_{\mathbb{T}^3}}^2 \right]. \tag{15}$$

This objective allows us to train the model without simulating the entire stochastic process, greatly enhancing computational efficiency. A critical concern when introducing stochasticity is the potential violation of fundamental physical symmetries. To enforce the conservation of chirality—a

crucial property of biomolecules—we introduce the chirality penalty term into the loss function (Ciampiconi et al., 2024). Let $M$ denote the operator that flips the chirality of a structure. We require the learned vector field to exhibit asymmetry under this transformation: $v_\theta(M(\chi_t), t) = -M_*(v_\theta(\chi_t, t))$. This is enforced via the chiral asymmetry loss:

$$\mathcal{L}_{\text{chiral}} = \mathbb{E}_{\chi_t, t} \left[ [\| v_\theta(M(\chi_t), t) + M_*(v_\theta(\chi_t, t)) \|^2_{g_{\mathbb{T}^3}}] \right]. \tag{16}$$

### 3.2.3 TOTAL OBJECTIVE AND SUMMARY OF IMPROVEMENTS

The overall training objective for the Stochastic Flow Matching part of ChiFlow combines the flow matching loss and the chiral symmetry loss:

$$\mathcal{L}_{\text{total}}^{\text{SFM}} = \mathcal{L}_{\text{SFM}} + \lambda \mathcal{L}_{\text{chiral}}. \tag{17}$$

The hyperparameter $\lambda$ balances the trade-off between sample diversity (achieved through SFM) and strict geometric consistency (enforced by the chiral loss) and is defined in the config file during training between 0.1 and 2. The controlled stochastic noise enables the generation of a wider variety of structurally plausible protein backbone. The SFM extension transforms ChiFlow into a more powerful and versatile tool for generating diverse, robust, and physically accurate molecular structures on the toroidal manifold.

### 3.3 C$\alpha$ COORDINATE RECONSTRUCTION

We employ forward kinematics with standard protein backbone parameters (e.g., C$\alpha$–N = 1.45 Å, C$\alpha$–C = 1.54 Å) to reconstruct C$\alpha$ coordinates from $\chi_1$ torsional angles. To maximize computational efficiency and numerical stability, we adopt the Self-Normalizing Natural Extension Reference Frame (SN-NeRF) algorithm Parsons et al. (2005). This method reduces the Cartesian coordinate conversion to 66 floating-point operations per atom, which is fewer than classical rigid-based reconstruction methods (Yim et al., 2023b). Angle-dependent placement: Compute initial atom position in a local frame using bond lengths and angles.

$$\vec{D}_2 = (R \cos \theta, \quad R \cos \phi \sin \theta, \quad R \sin \phi \sin \theta) \tag{18}$$

where $R = \text{bond}_{CD}$, $\theta = \text{angle}_{BCD}$, $\phi = \text{torsion}_{BC}$.

Reference frame alignment: Transform to global coordinates via an orthonormal basis derived from prior atoms:

$$\vec{D} = \hat{M}\vec{D}_2 + \vec{C} \tag{19}$$

$$\hat{M} \equiv \left[ \hat{bc}, \ \hat{n} \times \hat{bc}, \ \hat{n} \right] \tag{20}$$

with $\hat{bc} = \overrightarrow{BC}/|\overrightarrow{BC}|$ and $\hat{n}$ as the ABC plane normal.

Figure 5: SN-Nerf

## 4 EXPERIMENTS SETUP

### 4.1 DATASET

We train **ChiFlow** on a filtered PDB (Burley et al., 2022) dataset. The raw data was downloaded from the RCSB (Burley et al., 2022) in mmCIF format and we followed method proposed in FrameDiff (Yim et al., 2023b) to preprocess the dataset to pdb format for visualization and generated metadata.csv and clusters-by-entity-30.txt to help process the dataset more efficiently. During training, we use different length of protein backbone such as [60,256] and [100,512], as shown in Figure 7.Also, pair geometry is encoded via the features of protein backbone heavy atoms. What's more, during the preprocess process of the datasets, full in-memory caching is disabled by default to keep preprocessing lightweight.

### 4.2 METRIC

Designability, defined as the fraction of designs with scRMSD < 2.0 Å; Novelty, evaluated by the maximum TM-score to known PDB structures and the fraction of designs with an averaged

maximum TM-score $< 0.3$ and scRMSD $< 2.0$ Å, and Diversity, assessed by the average pairwise TM-score and the MaxCluster fraction. Standard errors are included for both Designability and Novelty metrics (Bose et al., 2024). And the Chirality was calculated through the Cahn-Ingold-Prelog (CIP) rule algorithm from RDKit (Parsons et al., 2005).

## 4.3 BASELINE

We compare our results to RFDiffusion (Watson et al., 2022), FrameDiff (Yim et al., 2023b), Frame-Flow (Yim et al., 2023a) and FoldFlow (Bose et al., 2024) for protein backbone generation. Unlike RFDiffusion which leverages a pre-trained folding network, both Genie and FrameDiff are diffusion models that generate protein backbones without such dependency. For the flow-based models, FrameFlow and FoldFlow, we utilize their official implementations and pre-trained weights, these baselines are selected to illustrate the inherent trade-offs between computational speed and design quality in state-of-the-art protein structure generation. We can tell from Table 1 that while there still remains a gap between the golden standard(RFDiffusion) and ChiFlow, ChiFlow approaches previous methods like FrameFlow, FrameDiff and FoldFlow in benchmark and shows no big difference. Note that the SFM greatly improved the Diversity of the protein backbone designed by ChiFlow, even outperform the golden standard. And Figure 7 shows some of the backbone generated by ChiFlow. The ablation study 2 shows that the Riemannian mirroring operator is crucial to the control of the Chirality of the protein backbone. And the Riemannian HyperTorus ensures that the Flow Matching process on dihedral angles is fluent and smooth. Overperform SE(3)-only Flow Matching method. **Note:** Best results are in bold. Standard errors represent variability across multiple runs. Chirality purity measures the percentage of generated backbones with correct L-chirality configuration.

Table 1: Comprehensive experiments of Baseline over the PDB Datasets Under Various Metrics

| Model | Designability | | Novelty | | Diversity | | Chirality |
|---|---|---|---|---|---|---|---|
| | Fraction ↑ | scRMSD ↓ | Fraction ↑ | avg.max TM ↓ | pairwise TM ↓ | MaxCluster ↑ | Score ↑ |
| RFDiffusion | **0.969** ± 0.034 | **0.650** ± 0.136 | **0.708** ± 0.060 | 0.449 ± 0.012 | 0.256 ± 0.010 | 0.172 ± 0.015 | 0.94 ± 0.143 |
| FrameDiff | 0.414 ± 0.064 | 3.970 ± 0.436 | 0.181 ± 0.128 | 0.556 ± 0.047 | 0.244 ± 0.011 | **0.320** ± 0.021 | 0.88 ± 0.094 |
| FrameFlow | 0.798 ± 0.044 | — | — | 0.698 ± 0.033 | — | 0.292 ± 0.020 | 0.93 ± 0.039 |
| FoldFlow | 0.671 ± 0.044 | 3.090 ± 0.285 | 0.449 ± 0.078 | 0.469 ± 0.025 | 0.274 ± 0.014 | — | 0.95 ± 0.047 |
| **ChiFlow** | 0.674 ± 0.042 | 2.309 ± 0.271 | 0.383 ± 0.074 | 0.502 ± 0.024 | 0.267 ± 0.013 | 0.295 ± 0.019 | **1.00** |
| **ChiFlowpro** | 0.743 ± 0.040 | 2.192 ± 0.249 | 0.497 ± 0.068 | **0.409** ± 0.020 | **0.197** ± 0.010 | 0.302 ± 0.020 | **1.00** |

Table 2: Comprehensive ablation study of ChiFlow components. The table evaluates the contribution of explicit chirality constraints and Stochastic Flow Matching (SFM) to the overall performance.

| Method | Designability | | Novelty | | Performance | |
|---|---|---|---|---|---|---|
| | Fraction ↑ | scRMSD (Å) ↓ | Fraction ↑ | Avg. TM ↓ | Diversity ↓ | Chirality (%) ↑ |
| **ChiFlow-pro (Full)** | **0.743** ± 0.040 | **2.192** ± 0.249 | 0.497 ± 0.068 | **0.409** ± 0.020 | **0.197** ± 0.010 | 1.00 |
| SE(3) | 0.671 ± 0.044 | 3.090 ± 0.285 | 0.449 ± 0.078 | 0.469 ± 0.025 | 0.274 ± 0.014 | 0.95 ± 0.047 |
| ChiFlow(w/o Chirality) | 0.735 ± 0.039 | 2.205 ± 0.251 | 0.505 ± 0.069 | 0.425 ± 0.021 | 0.275 ± 0.010 | 0.623 ± 0.057 |

**Configuration:** The model learns a continuous vector field over the backbone torsion angles $(\phi, \psi, \omega)$ in the 3-torus space $(-\pi, \pi]^3$, and reconstructs Cartesian coordinates $(N, C_\alpha, C)$ via a deterministic NeRF decoder. The system was trained for 500 epochs using Adam ($lr = 1 \times 10^{-4}$, $\beta_{default}$, batch size 512) with gradient norm clipping at 1.0. Distributed Data Parallel (DDP) (Li et al., 2020) on two A6000 GPUs was used for training, synchronizing gradients via an All-Reduce algorithm. Sampling integrates the learned field using 100 steps of explicit Euler reverse diffusion with torus wrapping, augmented by stochastic paths (SFM) to promote diversity. All runs were configured and versioned via Hydra. And inference is executed on a single RTX 4090 GPU (24 GB GDDR6X, 330 TFLOPS FP16).

## 5  RELATED WORKS

**Flow-based models.**   Flow models learn a time-dependent vector field and generate structures via ODE or controlled SDE integration. SE(3)-equivariant frame flows introduced by FrameFlow target backbone generation with fast sampling and strong designability (Yim et al., 2023a), followed by improvements in conditional motif-scaffolding (Yim et al., 2024) and added stochasticity for diversity (Bose et al., 2024). FoldFlow-base/OT/SFM refines flow-matching objectives and training, reporting improved designability and throughput across variants (Bose et al., 2024). In parallel, FoldFlow2 advances training stability and sampling efficiency through refined flow objectives and architectural updates, achieving further gains in both accuracy and designability (Huguet et al., 2024). More recently, PROTEINA extends flow-based generation with hierarchical representations and geometry-aware conditioning, enabling scalable modeling of complex protein families (Geffner et al., 2025). In general, flows sample deterministically, support direct conditioning in the vector field, and offer faster sampling than diffusion in comparable setups.

**Diffusion-based models.**   Diffusion models construct a stochastic forward noising process and learn a reverse denoising dynamics to sample structures. RFDiffusion popularized diffusion in protein backbone design, enabling flexible conditional generation and motif scaffolding (Watson et al., 2022). Subsequent motif-conditioned diffusion systems extended conditioning mechanisms for joint sequence–structure tasks (Song et al., 2023). Latent diffusion accelerates sampling by operating in an autoencoded Euclidean latent space (Fu et al., 2023), trading some designability for speed. Large-scale scaffold diffusion emphasizes dataset scale and diversity, improving novelty and coverage while retaining conditional control (Lin et al., 2024b). Diffusion models are versatile and robust but typically require more sampling steps and, when operating in SE(3) or Euclidean latents, do not align natively with the periodic geometry of dihedral angles.

**Chirality Constraints.**   Chirality, the property of a molecule being non-superimposable on its mirror image, is a critical factor in chemistry and pharmacology. Enantiomers, or pairs of chiral molecules, can exhibit vastly different biological activities; for instance, one enantiomer of a drug can be therapeutic while the other is toxic (Dumitrescu et al., 2025a; Gaiński et al., 2023). Consequently, the ability of computational models to accurately represent and predict chirality is of paramount importance, particularly in drug discovery (Gaiński et al., 2023). Historically viewed as a binary property, recent work has also focused on developing mathematical methods to quantify chirality as a continuous variable, with primary approaches based on either geometric overlap, such as the Continuous Chirality Measure (Grieder et al., 2025), or scalar triple products, like the Chirality Characteristic ($\chi$) (Abramson et al., 2024). Despite its importance, modeling chirality presents a significant challenge for many previous models such as SE(3)-based models (Dumitrescu et al., 2025a; Childs et al., 2025).

## 6  CONCLUSION

We present **ChiFlow**, a novel generative model for protein backbones that directly addresses the critical challenge of protein backbone chirality preservation—a fundamental constraint overlooked by existing SE(3)-based methods. By reformulating structure generation on a toroidal riemannian manifold and incorporating an asymmetry constraint, ChiFlow guarantees perfect chiral consistency without compromising performance. Furthermore, by operating natively on the space of torsion angles, ChiFlow avoids the computational overparameterization associated with SE(3) models and well models the dihedral angles of the protein backbone, resulting in a simpler architecture and faster sampling. Also, by utilizing the SFM, we improve the diversity of the protein backbone generated by ChiFlow. Despite these advantages, the torsional representation is inherently sensitive to local angular deviations, which can accumulate over very long proteins. Future work could focus on improving the stability of the generative process for extended chains, as well as exploring conditional generation for functional motifs and multimeric assemblies. ChiFlow establishes the toroidal manifold as an efficient way for protein backbone generation, offering a promising path toward chirality-aware design in computational structural biology.

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

# A    SUPPLEMENTARY PROOFS AND TECHNICAL DETAILS

## A.1    SCHRÖDINGER BRIDGE THEORY

The Schrödinger bridge problem (SBP) finds the most likely stochastic evolution between two probability distributions given a prior stochastic evolution. It is the dynamical version of the entropy-regularized optimal transport (OT) problem (Li et al., 2024), where the mean square distance is replaced by the relative entropy (Shi et al., 2025). Specifically, given a prior diffusion process $Q^\gamma$ (e.g., a Wiener measure with volatility $\gamma$) and two endpoint marginals $\pi_0$ and $\pi_1$, the dynamic SBP is defined as:

$$\inf_{Q \in \mathcal{D}(\pi_0, \pi_1)} D_{\mathrm{KL}}(Q \| Q^\gamma), \tag{21}$$

where $\mathcal{D}(\pi_0, \pi_1)$ denotes the set of measures on path space with marginals $\pi_0$ at time 0 and $\pi_1$ at time 1. The minimizing $Q^*$ is called the *Schrödinger bridge*, and its optimal value is the *entropic transportation cost*.

The SBP admits a dual formulation through the *Schrödinger system*, a pair of coupled PDEs (or integral equations in discrete settings) for potentials $\varphi$ and $\hat{\varphi}$. For a diffusion prior with drift $b_0$ and volatility $\sqrt{\gamma}$, the optimal drifts of the forward and backward Schrödinger bridges are given by:

$$b^+(t) = \gamma \nabla_x \varphi(x^+(t), t), \qquad b^-(t) = \gamma \nabla_x \hat{\varphi}(x^-(t), t), \tag{22}$$

where $x^+$ and $x^-$ are the forward and backward processes, respectively. The potentials satisfy the boundary conditions:

$$\varphi(x, 0) \propto \frac{\mathrm{d}\pi_0}{\mathrm{d}Q_0^\gamma}(x), \quad \hat{\varphi}(x, 1) \propto \frac{\mathrm{d}\pi_1}{\mathrm{d}Q_1^\gamma}(x), \tag{23}$$

with $Q_0^\gamma$ and $Q_1^\gamma$ being the marginals of the prior at $t = 0$ and $t = 1$.

A key numerical approach for solving the SBP is the *iterative proportional fitting procedure* (IPFP) (Eckstein & Lakhal, 2025), also known as the Sinkhorn algorithm in the static OT setting. For dynamic SBP, IPFP alternates between solving forward and half-bridge problems:

$$P_i = \arg \inf_{P \in \mathcal{D}(\cdot, \pi_1)} D_{\mathrm{KL}}(P \| Q_{i-1}), \tag{24}$$

$$Q_i = \arg \inf_{Q \in \mathcal{D}(\pi_0, \cdot)} D_{\mathrm{KL}}(Q \| P_i), \tag{25}$$

which converges to the Schrödinger bridge $Q^*$. This procedure is equivalent to an iterative scaling algorithm in the discrete case.

Recently, (Teter et al., 2024) proposed a *regularized* SBP variant with a quadratic state cost-to-go, which incentivizes paths to stay close to a nominal level. Unlike the conventional SBP, this regularization induces a state-dependent rate of killing and creation of probability mass. Remarkably, they showed that this regularized SBP is *exactly solvable* even for non-Gaussian endpoints, by deriving the Markov kernel of the associated reaction-diffusion PDE in closed form. Their solution recovers the heat kernel (and hence the conventional Schrödinger bridge) as a special case when the regularization vanishes.

The SBP also extends to quantum channels, where the goal is to scale a completely positive operator $Q$ to a quantum channel $R$ such that $R(\alpha) = \beta$ for given density matrices $\alpha$ and $\beta$. This is the quantum analog of the classical SBP and can be solved using fixed-point methods.

## A.2    SE(3)-INVARIANT FEATURE DISTRIBUTIONS CANNOT ENCODE CHIRALITY

We formalize why standard E(3) / SE(3)-invariant (or equivariant) architectures based purely on Euclidean distances and inner products cannot distinguish mirror-image molecular or protein backbone configurations. This limitation motivates our explicit asymmetry constraint on dihedral torus flows.

**Proposition A.1** (Inability of E(3)-invariant point-cloud distributions to distinguish enantiomers)**.** *Let $p_\phi$ be a distribution over point clouds $M = (m_1, \ldots, m_N) \in (\mathbb{R}^3)^N$ parameterized only through features that are invariant under the action of the Euclidean group E(3) (translations +*

rotations) and depend exclusively on pairwise distances $\|m_i - m_j\|$ and/or inner products of centered coordinates. Then for any enantiomeric pair $(M, M')$ related by an improper orthogonal transformation (a reflection), we have

$$p_\phi(M) = p_\phi(M'). \tag{26}$$

Consequently, $p_\phi$ cannot enforce or prefer a single molecular chirality.

**Sketch of proof.** Any collection of features constructed from (i) pairwise Euclidean distances $d_{ij} = \|m_i - m_j\|$, (ii) dot products of centered vectors $\langle m_i - \bar{m}, m_j - \bar{m} \rangle$, or (iii) higher-order tensor contractions thereof is invariant under the full orthogonal group O(3), because both distances and inner products are preserved by any orthogonal matrix $Q$ with $Q^T Q = I$, including reflections ($\det Q = -1$). A reflection that maps $M$ to its mirror image $M'$ lies in O(3)$\backslash$ SO(3). Thus all such features take identical values on $M$ and $M'$ (Dumitrescu et al., 2025b). strengthen this by showing that for functions of at most $n$ input vectors, SE($n$)-invariance already implies reflection invariance via centering arguments and Householder reflections; hence SE(3)- invariant parameterizations cannot break mirror symmetry. Therefore any likelihood or density model $p_\phi$ expressed purely in those invariants assigns equal probability to both enantiomers.

**Implication for ChiFlow.** To obtain chirality selectivity, one must either (a) augment invariant features with orientation-sensitive pseudoscalars (e.g., triple products) or (b) impose explicit asymmetric constraints under a mirroring isometry, as we do via the torsional mirroring operator $M$ and the chiral penalty (Section 2.4).

### A.3 PROOF OF PROPOSITION 3.1 (CONDITIONAL DRIFT UNDER TORUS INTERPOLATION)

We give a concise proof for the closed form of the conditional target drift on the torus under the time-dependent affine interpolation used in ChiFlow. Let $\tau \in \mathbb{T}^d$ denote a generic angle vector (componentwise on the circle), and consider an interpolation of the form

$$\tau_t = (\mu_t(\tau_1) + \sigma_t(\tau_1)\varepsilon) \bmod 2\pi, \qquad t \in [0, 1], \tag{27}$$

where $\mu_t$ and $\sigma_t > 0$ are smooth in $t$ (and may depend on the endpoint $\tau_1$), and $\varepsilon$ is time-invariant along the conditional path. Differentiating $\tau_t$ w.r.t. $t$ in the tangent yields

$$u_t(\tau \mid \tau_1) = \frac{\mathrm{d}}{\mathrm{d}t}\tau_t = \dot{\sigma}_t\varepsilon + \dot{\mu}_t. \tag{28}$$

By rearranging the interpolation, we have the (componentwise, principal-branch) identity on the circle

$$\varepsilon = \frac{\mathrm{wrap}(\tau_t - \mu_t)}{\sigma_t}, \tag{29}$$

so substituting eliminates $\varepsilon$ and expresses the drift as a function of $(\tau, t)$:

$$u_t(\tau \mid \tau_1) = \dot{\sigma}_t\frac{\mathrm{wrap}(\tau_t - \mu_t)}{\sigma_t} + \dot{\mu}_t = \frac{\dot{\sigma}_t(\tau_1)}{\sigma_t(\tau_1)}\mathrm{wrap}(\tau - \mu_t(\tau_1)) + \dot{\mu}_t(\tau_1), \tag{30}$$

where all operations are taken componentwise on $\mathbb{T}^d$ using the shortest-arc difference $\mathrm{wrap}(\cdot) \in (-\pi, \pi]$. This is the claimed expression.

### A.4 PROOF OF PROPOSITION 3.2 (UNBIASEDNESS VIA DISINTEGRATION ON $\mathbb{T}^d$)

We show the equivalence between the conditional objective and its marginal counterpart by disintegration of measures on the product torus and Fubini's theorem (Rosestolato, 2018). Let $\mathbb{T}^d = S^1 \times \cdots \times S^1$ and assume the endpoint distributions $p_0, p_1 \in \mathcal{P}(\mathbb{T}^d)$ factorize across angles (orthogonality/independence of torsions), hence the interpolant $p_t$ admits the same disintegration. Define

$$u_t(\tau) = \mathbb{E}_{\tau_0 \sim p_0, \tau_1 \sim p_1}\left[\frac{p_t(\tau \mid \tau_0, \tau_1)}{p_t(\tau)}u_t(\tau \mid \tau_0, \tau_1)\right], \tag{31}$$

and consider the objective difference

$$\nabla_\theta\left(\mathbb{E}_{\tau_0, \tau_1, \tau \sim p_t(\cdot \mid \tau_0, \tau_1)}\left[\|v_t(\tau) - u_t(\tau \mid \tau_0, \tau_1)\|_g^2\right] - \mathbb{E}_{\tau \sim p_t}\left[\|v_t(\tau) - u_t(\tau)\|_g^2\right]\right), \tag{32}$$

where $\langle \cdot, \cdot \rangle_g$ and $\| \cdot \|_g$ denote the Riemannian inner product and norm induced by the product metric $g$ on $\mathbb{T}^d$. Expanding squares and canceling like terms reduces this to

$$-2 \nabla_\theta \Big( \mathbb{E}_{\tau_0, \tau_1, \tau \sim p_t(\cdot | \tau_0, \tau_1)} \langle v_t(\tau), u_t(\tau \mid \tau_0, \tau_1) \rangle_g - \mathbb{E}_{\tau \sim p_t} \langle v_t(\tau), u_t(\tau) \rangle_g \Big). \tag{33}$$

Using the definition of $u_t(\tau)$ and changing the order of integration,

$$\mathbb{E}_{\tau \sim p_t} \langle v_t(\tau), u_t(\tau) \rangle_g = \int \langle v_t(\tau), u_t(\tau) \rangle_g \, p_t(\tau) \, \mathrm{d}\tau \tag{34}$$

$$= \int \Big\langle v_t(\tau), \mathbb{E}_{\tau_0, \tau_1} \Big[ \tfrac{p_t(\tau | \tau_0, \tau_1)}{p_t(\tau)} u_t(\tau \mid \tau_0, \tau_1) \Big] \Big\rangle_g \, p_t(\tau) \, \mathrm{d}\tau \tag{35}$$

$$= \iint \Big\langle v_t(\tau), u_t(\tau \mid \tau_0, \tau_1) \Big\rangle_g \, p_t(\tau \mid \tau_0, \tau_1) \, p_0(\tau_0) \, p_1(\tau_1) \, \mathrm{d}\tau \, \mathrm{d}\tau_0 \, \mathrm{d}\tau_1 \tag{36}$$

$$= \mathbb{E}_{\tau_0 \sim p_0, \, \tau_1 \sim p_1, \, \tau \sim p_t(\cdot | \tau_0, \tau_1)} \langle v_t(\tau), u_t(\tau \mid \tau_0, \tau_1) \rangle_g. \tag{37}$$

Hence the two expectations are equal, and the gradient of their difference is zero. This establishes that optimizing the conditional objective is an unbiased surrogate for the marginal objective on $\mathbb{T}^d$ under the stated disintegration assumptions.

## A.5 PROOF THAT THE MIRRORING OPERATOR $M$ IS AN ISOMETRIC DIFFEOMORPHISM ON $\mathbb{T}^3$

We show that the mirroring map $M : \mathbb{T}^3 \to \mathbb{T}^3$ defined componentwise by

$$M(\phi, \psi, \omega) \;=\; (2\pi - \phi \bmod 2\pi, \; 2\pi - \psi \bmod 2\pi, \; \omega) \tag{38}$$

is an isometric diffeomorphism under the conformal product metric introduced in Section 2 (the extension to $\mathbb{T}^{3N}$ follows by the product structure, since $M$ acts residuewise).

**1) $M$ is a diffeomorphism.** Smoothness: Each component of $M$ is the composition of a smooth linear map on the circle (e.g., $\phi \mapsto 2\pi - \phi$) with the canonical identification modulo $2\pi$ that defines the smooth structure of the torus. Hence $M$ is smooth. Invertibility: $M$ is an involution, i.e., $M \circ M = \mathrm{id}$, so $M^{-1} = M$. Since $M$ is smooth, so is $M^{-1}$. Therefore $M$ is a diffeomorphism on $\mathbb{T}^3$; by independence across residues, the product map on $\mathbb{T}^{3N}$ is also a diffeomorphism.

**2) $M$ is an isometry for the metric $g_{\mathbb{T}^3}$.** Recall the metric

$$g_{\mathbb{T}^3}\big|_\chi(u, v) \;=\; \sum_{k \in \{\phi, \psi, \omega\}} w_k(\chi_k) \, u_k \, v_k, \qquad w_k(\chi_k) = \frac{1}{1 + \sin^2(\chi_k/2)}. \tag{39}$$

Let $\chi' = M(\chi)$ and let $M_* : T_\chi \mathbb{T}^3 \to T_{\chi'} \mathbb{T}^3$ be the pushforward of $M$. By differentiation of the component maps, the Jacobian of $M$ is $J_M = \mathrm{diag}(-1, -1, 1)$, hence for any $u \in T_\chi \mathbb{T}^3$ we have $M_* u = J_M u$ with components $(M_* u)_\phi = -u_\phi$, $(M_* u)_\psi = -u_\psi$, $(M_* u)_\omega = u_\omega$. Using the trigonometric identity $\sin(\pi - \theta) = \sin \theta$, we obtain for $k \in \{\phi, \psi\}$

$$w_k(\chi'_k) \;=\; \frac{1}{1 + \sin^2((2\pi - \chi_k)/2)} \;=\; \frac{1}{1 + \sin^2(\chi_k/2)} \;=\; w_k(\chi_k), \quad \text{and} \quad w_\omega(\chi'_\omega) = w_\omega(\chi_\omega). \tag{40}$$

Therefore, for any $u, v \in T_\chi \mathbb{T}^3$,

$$g_{\mathbb{T}^3}\big|_{\chi'}(M_* u, M_* v) \;=\; \sum_{k \in \{\phi, \psi, \omega\}} w_k(\chi'_k) \, (M_* u)_k \, (M_* v)_k \tag{41}$$

$$= w_\phi(\chi'_\phi)(-u_\phi)(-v_\phi) + w_\psi(\chi'_\psi)(-u_\psi)(-v_\psi) + w_\omega(\chi'_\omega) u_\omega v_\omega \tag{42}$$

$$= w_\phi(\chi_\phi) u_\phi v_\phi + w_\psi(\chi_\psi) u_\psi v_\psi + w_\omega(\chi_\omega) u_\omega v_\omega \tag{43}$$

$$= g_{\mathbb{T}^3}\big|_\chi(u, v). \tag{44}$$

which proves that $M$ preserves the metric, hence is an isometry. The product argument yields the same conclusion on $\mathbb{T}^{3N}$.

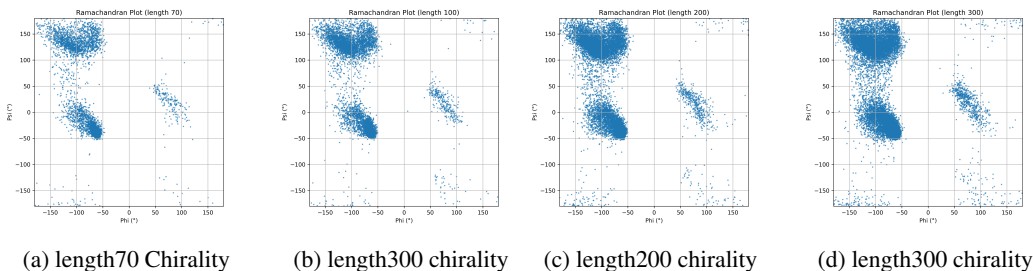

(a) length70 Chirality     (b) length300 chirality     (c) length200 chirality     (d) length300 chirality

Figure 6: Ramachandran distribution image of the generated backbone of different length.

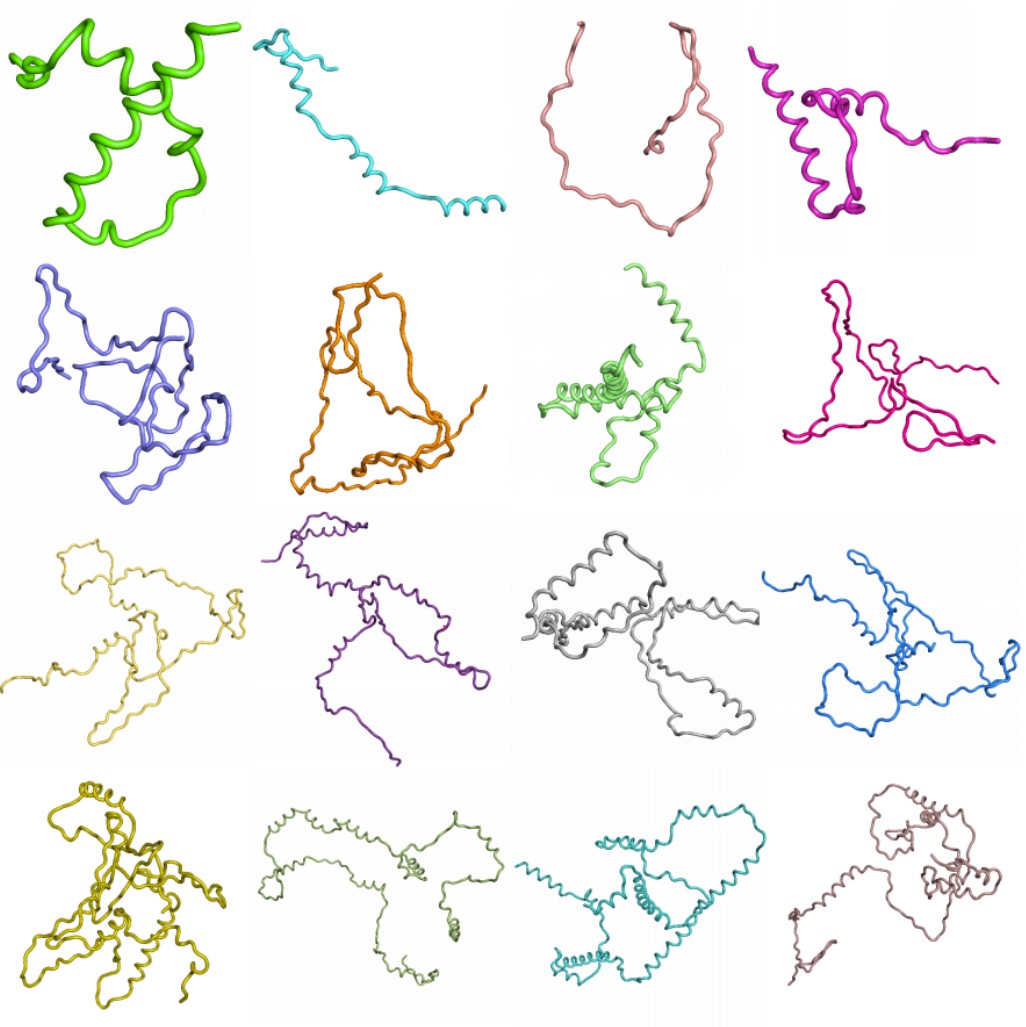

Figure 7: Different length of Protein Backbone Designed by ChiFlow. Varing from 60 to 256.

**Consequence.** Since $M$ is an isometric diffeomorphism and an involution, the asymmetry constraint $\hat{v}(M\chi, t) = -M_*\hat{v}(\chi, t)$ is metric-consistent and well-defined everywhere on $\mathbb{T}^{3N}$.

## A.6 THE USE OF LARGE LANGUAGE MODELS

In this paper, we used Deepseek to polish our writing and made use of copilot to help debug our code.

