# OpenReview forum: "ChiFlow: Torsional Asymmetry Flow Matching For Chirality-Aware Protein Backbone Generation"
_ICLR.cc/2026/Conference — ICLR 2026 Conference Withdrawn Submission_

### Official Review · Reviewer_LNFf · 2025-10-27

**Soundness:** 2
**Presentation:** 2
**Contribution:** 2
**Rating:** 2
**Confidence:** 4

**Summary:**

The authors propose ChiFlow, a toroidal flow matching framework that directly models torsional angles on the manifold $T^3$ and introduces an explicit mechanism to break mirror symmetry in the learned vector field, ensuring L-chiral conformations.

It provides a clean, theoretically grounded formulation of protein backbone generation by learning flows on the torus and embedding chirality asymmetry directly into the vector field—conceptually elegant and empirically solid, though still behind the strongest baselines in fine-grained designability and long-chain stability.

**Strengths:**

1. The training imposes an anti-symmetry constraint on the vector field, thereby forcing the learned flow to occupy only the physically valid L-chiral submanifold.
2. The authors extend the deterministic flow matching to a Stochastic Flow Matching (SFM) formulation on the torus, introducing a toroidal Brownian reference process governed by the Laplace–Beltrami operator. By reinterpreting SFM as a Schrödinger bridge problem, they improve sample diversity without violating chirality constraints.

**Weaknesses:**

1.  The contribution of the paper is rather limited. The core novelty mainly resides in three aspects:
	- 1).	the application of flow matching on a toroidal manifold for generative modeling of backbone torsional angles,
	- 2).	the explicit chirality enforcement mechanism to ensure the generation of valid L-chiral protein backbones,
	- 3).	a stochastic flow matching (SFM) extension to improve sampling diversity.

While each component is conceptually reasonable, none represents a fundamentally new methodological advance beyond existing flow matching or manifold-based generative frameworks. The toroidal formulation is a straightforward adaptation of Riemannian flow matching to periodic coordinates which has been explored by PPFlow, PocketFlow and etc., and the chirality constraint is an elegant but relatively simple antisymmetry regularization. Therefore, the overall contribution appears incremental rather than transformative.

2.The experimental validation is not sufficiently comprehensive, and the observed performance gains are relatively limited. Although the paper compares ChiFlow against several SE(3)-based baselines such as RFDiffusion, FrameFlow, and FoldFlow, the evaluation focuses primarily on backbone-level reconstruction metrics (e.g., scRMSD, designability, and chirality purity). However, there is a lack of functional or structural validation, such as fold-class conditional generation or long-chain generation tasks as proposed in Proteina. Conducting only standard and relatively simple experiments does not convincingly demonstrate the strengths or overall capabilities of the model — such as its scalability or functionality-guided generation potential.

Moreover, the quantitative improvements over baselines are marginal. The proposed approach actually performs worse than several state-of-the-art methods, with the only notable advantage being chirality purity — an unsurprising result given the explicit enforcement mechanism. In addition, the paper does not include a thorough comparison with other torsion-based generative models (e.g., FoldDiff), making it difficult to isolate and evaluate the true benefits of using flow matching or chirality constraints. As a result, the experimental section does not convincingly substantiate the claimed advantages of the proposed method.

3. The literature review is not sufficiently comprehensive. In line 80 and abstract, the authors claim that “unlike sidechain-focused torus approaches (Lin et al., 2024a)”, their work uniquely applies toroidal flow matching to backbone torsional angles. However, as far as I know, PPFlow also models torus-based flows directly on backbone torsion angles $(ϕ, ψ, and ω)$. This prior work should be properly discussed and cited, as it addresses a very similar problem formulation. Without acknowledging PPFlow and related torsion-space generative models, the paper’s claimed novelty is overstated and may give an inaccurate impression of the existing landscape.

4. Minor Issues and Typos
	-	Line 125: should use \cite{} instead of \citet{} for proper citation formatting.
	-	Line 126: should use \citep{} instead of \cite{} to maintain consistent reference style.
	-	Equation (15): the comma after the equation should be removed.
and etc. The paper seems not mature enough to be published.

**Questions:**

1.	Novelty of Flow Matching on the Torus
The proposed flow matching on the torus appears methodologically identical to PPFlow, which also performs flow matching directly on backbone torsion angles $(ϕ, ψ, ω)$. Given that PPFlow already formulates backbone generative modeling in a torus manifold, it is unclear what specific methodological innovation ChiFlow introduces beyond that prior work. The authors are encouraged to clarify the conceptual or algorithmic differences relative to PPFlow — for instance, whether the improvements lie in training objectives, vector field parameterization — to substantiate the claimed novelty.

	2.	Necessity of Explicit Chirality Modeling
As shown in the ablation study, SE(3)-based baselines such as FoldFlow already achieve around 0.95 chirality purity without any explicit constraint. This raises the question of whether chirality violation is truly a significant issue in practice. Given that the proposed torsion-space modeling substantially degrades RMSD and designability performance compared to stronger methods like RFDiffusion and Proteina, is it worthwhile to sacrifice structural accuracy for marginal improvements in chirality purity? A clearer justification or quantitative trade-off analysis would be valuable for assessing the real benefit of introducing the chirality constraint.

3. Presentation and Focus
The paper contains substantial redundancy in the methodological exposition. Sections describing conditional Riemannian flow matching and the NeRF reconstruction algorithm repeat well-established concepts that have already been thoroughly introduced in previous literature.

To improve the paper’s focus and balance, it would be advisable to streamline or shorten these background sections, and instead allocate the saved space to additional experimental validation. For example, incorporating experiments similar to those in Proteina—such as function-conditioned generation or scaling to longer chains—would significantly enhance the depth, credibility, and overall impact of the experimental evaluation. This shift would help demonstrate the practical advantages of ChiFlow more convincingly than reiterating well-known methodological details.

---

### Official Review · Reviewer_ByCH · 2025-10-31

**Soundness:** 3
**Presentation:** 3
**Contribution:** 2
**Rating:** 4
**Confidence:** 4

**Summary:**

This paper introduces **ChiFlow**, a *chirality-aware flow-matching framework* for protein backbone generation. Unlike existing **SE(3)-based** models such as FrameFlow and FoldFlow2, ChiFlow models backbone dihedral angles \((\phi, \psi, \omega)\) directly on a **toroidal Riemannian manifold** \(T^3\), addressing two key limitations of prior work: (1) periodicity artifacts caused by embedding cyclic angles in SE(3), and (2) lack of explicit chirality control.

The method introduces three main contributions:
1. A **toroidal Riemannian manifold formulation** that natively handles cyclic dihedral angles.
2. A **Riemannian mirroring operator** and asymmetric vector field constraint that enforce *L-chirality*.
3. A **Stochastic Flow Matching (SFM)** extension that improves conformational diversity while preserving chirality.

Experiments on PDB-derived datasets show that ChiFlow achieves **perfect chirality purity (100%)** and comparable designability and novelty to SE(3)-based baselines, while also improving structural diversity.

**Strengths:**

Introducing an explicit chirality constraint fills a gap in existing SE(3)-based generative models, which cannot inherently distinguish mirror-symmetric configurations. The proposed design could serve as a foundation for future research on *chiral molecule* and *protein generative models*.

**Weaknesses:**

1. Using torsion-based generative models for protein generation is not a new concept, e.g. FoldinfDiff (Protein structure generation via folding diffusion).

2. Using flow matching for protein side-chain or torsion angle generation is also not a new approach (e.g. PepFlow, FlowPacker).

**Questions:**

1. Using flow matching and torsion flow matching is not a new concept; I encourage authors to get more related work and explain the difference between ChiFlow and previous work.
2. How does ChiFlow’s computational efficiency compare to FrameFlow and FoldFlow in both training and inference?
3. Could additional ablation studies be included (e.g., varying the chirality loss weight \(\lambda\), removing SFM)?
4. Can the method handle large protein complexes or asymmetric protein complex generation?

---

### Official Review · Reviewer_q6cy · 2025-11-05

**Soundness:** 3
**Presentation:** 2
**Contribution:** 4
**Rating:** 6
**Confidence:** 2

**Summary:**

The paper proposes ChiFlow, a protein backbone generator that models the three backbone dihedrals (ϕ, ψ, ω) directly on a toroidal Riemannian manifold and trains with (stochastic) flow matching. The key idea is to avoid SE(3) over-parameterization for cyclic angles and to enforce L-chirality by introducing a mirroring operator and an asymmetry constraint on the learned vector field. The model reconstructs N–Cα–C coordinates deterministically from dihedrals via SN-NeRF. On a filtered PDB dataset, ChiFlow reports competitive designability and diversity relative to recent flow baselines while achieving 100% chirality purity.

**Strengths:**

- The paper presents a clear geometric motivation by modeling torsion angles on a toroidal manifold, effectively avoiding boundary artifacts and overparameterization that occur in SE(3)-based models.
- It introduces a simple yet mathematically consistent chirality constraint using a mirroring operator that ensures all generated backbones are strictly L-chiral throughout training.
- Empirical evaluations demonstrate that ChiFlow achieves 100% chirality purity and approaches or surpasses flow baselines in diversity and novelty while retaining strong designability scores.
- The authors provide detailed training setups, ablations, and dataset preprocessing steps, improving reproducibility.

**Weaknesses:**

- The evaluation focuses mainly on unconditional backbone generation and lacks discussion/experiments on conditional or motif-scaffolding tasks. In real life, conditioned generation seems to be more important for practical protein design applications.
- Although the paper highlights chirality enforcement, it does not address full-atom reconstruction or side-chain modeling, leaving open questions about usability for sequence design and structure-function validation.
- While the paper acknowledges that torsional representation may cause instability for long proteins, no robustness analysis or extended-length tests are provided.
- Writing quality is inconsistent, with grammatical errors and ambiguous phrasing that obscure technical clarity and may hinder readability. (e.g., "comparasion" in Line 41)

**Questions:**

(See above)

---

### Official Review · Reviewer_Rfrg · 2025-11-07

**Soundness:** 2
**Presentation:** 4
**Contribution:** 2
**Rating:** 2
**Confidence:** 4

**Summary:**

A protein design method is proposed that models the protein backbone geometry by triplets of dihedral angles. Each triplet modeled as a point on the hypertorus $T^3$. The whole protein's configuration of dihedral angles then is modeled as the manifold of n hypertoruses $T^{3n}$. The authors propose a consistency constraint ("mirroring operator") of the dihedral angles that ensures a consistent chirality along the proteins backbone. For sampling a protein structure, a Riemannian flow matching approach is derived that samples a configuration of the dihedral angles on the manifold $T^{3n}$.

**Strengths:**

The mathematical presentation of the Riemannian flow matching approach is excellent. It is delightful to read such a well presented mathematical derivation in a machine learning paper. The mathematical derivations seem to be sound and the references are detailed.

**Weaknesses:**

I have doubts about the significance of the contribution of this paper. The authors motivate the proposed method by its unique property, that chirality can be predefined and preserved. This would be in contrast to existing protein design methods that use a SE(3) or E(3) representation of the protein.
However, it is unclear if the chirality metric in table 1 would include proteins with consistent D-chirality for which the correct L-chirality could be recovered by simple mirroring.

High designability of RFDiffusion seems to be based on a high amount of alpha-helices in the protein samples. A bias that is being explored in recent literature, e.g. Wagner et. al. 2024. Please additionally report alpha helix and beta strand content for ChiFlow.

The use of euclidean coordinate representations of the protein backbone has contributed to the breakthroughs in computational methods for protein folding, e.g. Alphafold, and protein design. The method proposed in this paper seems to go back to a kinematic chain, with all the problems that might arise by this. This should be properly motivated and its benefits need to be demonstrated by rigorous experiments.


Technical remarks:

"where ut(x|x1) generates pt(x|x1) from p0." - This sentence requires further explanation as all quantities are intdependent of p0.

Dumitrescu et al. 2025b, E(3)-equivariant models cannot learn chirality, seems improperly cited: "SE(3) which can not model chirality". Here it is important to make a distinction between E(3) and SE(3).

References are missing journal / conference names. E.g. the citation for Mathieu and Nickel 2020 appears as
Emile Mathieu and Maximilian Nickel. Riemannian continuous normalizing flows, December 2020.
however, it should be
Emile Mathieu and Maximilian Nickel. "Riemannian continuous normalizing flows." Advances in neural information processing systems 33 (2020): 2503-2515.


References:
Wagner, S., Seute, L., Viliuga, V., Wolf, N., Gräter, F., & Stühmer, J. (2024). Generating highly designable proteins with geometric algebra flow matching. Advances in Neural Information Processing Systems, 37, 77987-78026.

**Questions:**

How exactly is the Chirality metric computed? ("Chirality purity measures the percentage of generated backbones with correct L-chirality configuration.")

How many of the samples of the baselines with incorrent L-chirality could be corrected by simply mirroring them?

How would the Chirality score in table 1 change if we would also include these samples that would be correct after mirroring?

---

### Official Review · Reviewer_tq1J · 2025-11-08

**Soundness:** 1
**Presentation:** 1
**Contribution:** 2
**Rating:** 2
**Confidence:** 4

**Summary:**

This paper proposes ChiFlow, a chirality-aware torsional asymmetric flow matching for unconditional protein backbone generation. They model backbones with three dihedrals $\phi$, $\psi$, $\omega$ as points on $\mathbb{T}^3$ and reconstruct using the SN-NeRF algorithm. They also introduce a loss based on the Riemannian mirroring operator to encourage chirality purity.

**Strengths:**

This work introduces a new paradigm for protein backbone modeling centered on dihedrals, offering an alternative to SE(3) frame or coordinate-based methods. Experimental results demonstrate that the ChiFlow achieves 100% L-chirality.

**Weaknesses:**

**W1**.This paper presents a fundamental technical inconsistency: it theoretically claims to use a complex non-flat Riemannian metric (Eq. 8), but the geodesic path (Eq. 10) and vector field (Eq. 12) it uses for the implementation are only valid for a flat metric.

**W2**.This paper claims to guarantee absolute L-chirality, but the mechanism for this is a chiral asymmetry loss $\mathcal{L}_{\text{chiral}}$, which is regarded as a kind of soft constraint. This is not a formal, architectural enforcement that provides a mathematical guarantee.

**W3**.The presentation of the experiment part is not clear. For example:

(a). Protein lengths, numbers generated for the evaluation are not mentioned.

(b). The definition of ChiFlowpro is not explicitly explained.

(c). The results analysis is presented in the 'Baseline' subsection.

**W4**. The paper's claim of '...while uniquely delivering one hundred percent chirality purity, bridging a fundamental gap in backbone generation.' (line 82-84)  appears to be an overstatement. While ChiFlow achieves 100% L-chirality purity, the gap is not as fundamental as implied, e.g.,  FrameFlow is 93% and RFDiffusion is 94%.

**Questions:**

**Q1**. Since previous works, e.g., DiffPack[1], FlowPacker[2], have used dihedrals for side-chain generation, and ChiFlow uses dihedrals for backbone generation, would it be possible to unify them into a single framework for full-atom protein generation?

[1].DiffPack: A Torsional Diffusion Model for Autoregressive Protein Side-Chain Packing

[2].FlowPacker: Protein side-chain packing with torsional flow matching

---

### Note · Authors · 2025-11-18

**Comment:**

# Subject: Withdrawal of Manuscript 6786

Dear ICLR Editors and Reviewers,

Thank you for providing the constructive review comments on our manuscript titled "ChiFlow: Torsional Asymmetry Flow Matching For Chirality-Aware Protein Backbone Generation"
We have carefully reviewed all the feedback and agree that the paper requires significant revisions to meet the high standards of the conference. After internal discussion, we have decided to withdraw the current version​ of the manuscript.
We sincerely appreciate the reviewers' insightful suggestions, which will help us greatly in improving the work. We plan to refine the paper thoroughly and plan to resubmit it in the future when it is better developed.
Thank you again for your time and effort!

Sincerely,

Authors of submission 6786

**Withdrawal Confirmation:**

I have read and agree with the venue's withdrawal policy on behalf of myself and my co-authors.